# Usefulness of Muscle Ultrasound to Study Sarcopenic Obesity: A Pilot Case-Control Study

**DOI:** 10.3390/jcm11102886

**Published:** 2022-05-19

**Authors:** Andreu Simó-Servat, Montse Ibarra, Mireia Libran, Silvia Rodríguez, Verónica Perea, Carmen Quirós, Aida Orois, Noelia Pérez, Rafael Simó, Maria-José Barahona

**Affiliations:** 1Department of Endocrinology and Nutrition, Hospital Universitari Mútua Terrassa, Plaça del Doctor Robert, 5, 08221 Terrassa, Spain; mibarra@mutuaterrassa.cay (M.I.); mlibran@mutuaterrassa.es (M.L.); sirodriguez@mutuaterrassa.cat (S.R.); vperea@mutuaterrassa.cat (V.P.); cquiros@mutuaterrassa.cat (C.Q.); aorois@mutuaterrassa.cat (A.O.); 2Department of Medicine, Autonomous University of Barcelona, 08193 Bellaterra, Spain; 3Department of General Surgery, Hospital Universitari Mútua Terrassa, 08221 Terrassa, Spain; 35667npr@gmail.com; 4Diabetes and Metabolism Research Unit, Vall d’Hebron Research Institute and CIBERDEM (ISCIII), Autonomous University of Barcelona, 08193 Bellaterra, Spain; rafael.simo@vhir.org

**Keywords:** sarcopenic obesity, fat mass, lean mass, ultrasound

## Abstract

Background and objectives: Sarcopenic obesity (SO) is an emerging problem, especially in candidates for bariatric surgery (BS). We hypothesized that musculoskeletal ultrasound (MUS), a simple and accessible method, could be a reliable index of SO. Materials and Methods: A cross-sectional pilot study including 122 subjects (90 cases and 32 controls, 73% female, mean age: 51.2 years) who underwent BS was conducted at University Hospital Mútua Terrassa. The lean mass (LM) was calculated by bioelectrical impedance analysis (BIA) and the thigh muscle thickness (TMT) by MUS. To identify the subjects with SO by BIA, we used skeletal muscle index (SMI). The validity of MUS was determined using the ROC curve. Results: The mean BMI in the obesity group was 44.22 kg/m^2^. We observed a correlation between the LM and SMI assessed by BIA and the TMT assessed by MUS (R = 0.46, *p* < 0.001). This correlation was maintained at significant levels in the SO group (*n* = 40): R = 0.79; *p* = 0.003). The TMT assessed by MUS was able to predict SMI using BIA (AUC 0.77; 95% CI: 0.68242 to 0.84281). The optimal cut-off point for maximum efficiency was 1.57 cm in TMT (sensitivity = 75.6% and specificity = 71.1%). Conclusions: The TMT of the quadriceps assessed by US is a useful tool for identifying subjects with SO. Larger studies to validate this simple low-cost screening strategy are warranted.

## 1. Introduction

Although there are different definitions of sarcopenic obesity (SO) and its diagnostic criteria and cut-offs are not universally established, this entity can cause metabolic disorders and physical disability [1,2] and is characterized by the combination of low muscle mass and strength with increased fat mass (FM) [3,4,5]. In the latest consensus by the European Society for Clinical Nutrition and Metabolism (ESPEN) and the European Association for the Study of Obesity (EASO), SO was defined as the coexistence of excess adiposity and low muscle/mass function [6,7].

Loss of muscle mass and function is not exclusive of the elderly and is commonly accompanied by relative or absolute body fat gain. Therefore, sarcopenia may arise in individuals with obesity at any age, and it can not only significantly affect health outcomes, but also influence the evolution after bariatric surgery (BS) [1,7]. There is emerging evidence indicating that SO may be associated with an increased risk of global mortality and cardiovascular risk factors compared to sarcopenia or obesity alone [8,9]. In fact, ESPEN and EASO have recently recognized SO as a priority for clinical research [10,11]. Therefore, there is a growing interest in having accessible imaging techniques to detect and monitor SO. Furthermore, the study of body composition has several advantages over the mere assessment of BMI in obese subjects [3].

In order to assess muscle mass loss, computed tomography (CT), magnetic resonance imaging (MRI), dual-energy X-ray absorptiometry (DXA), and bioelectrical impedance analysis (BIA) can be used. DXA, along with CT and MRI, which are considered the gold standard, is difficult to implement in routine clinical practice due to its increased cost and demand for specialized professionals. Moreover, DXA needs space and has subject-related limitations (maximum weight 160 kg). By contrast, BIA is a relatively simple, quick, non-invasive and non-expensive method for assessing body composition, but has been shown to slightly overestimate fat-free mass in obese subjects [1,12,13,14]. Although some evidence suggests that musculoskeletal ultrasound (MUS) could be useful for quantifying body composition, and in particular lean mass (LM) in healthy adults, there is no information on this issue in patients with obesity [15]. MUS is a rapid, inexpensive, non-invasive diagnostic tool, without radiation, that can be performed in every clinical setting: community and hospital [16,17]. 

We hypothesized that MUS would allow a reliable assessment of sarcopenia in obese patients and could be used as a cost-effective alternative to the conventionally used skeletal muscle mass index (SMI) measured by BIA or DXA [18], as a screening proof. In addition, MUS could detect site-specific muscle loss that may appear in early stages of sarcopenia [19,20]. On this basis, the aim of this pilot study was to compare the thigh muscle thickness (TMT) measurement obtained by MUS with that obtained by using BIA as a conventional method. This approach will permit us to gain new insights on the potential usefulness of MUS as a tool for assessing SMI and, therefore, for identifying SO in the obese population and deciding which patients need gold standard techniques to determine sarcopenia. 

## 2. Materials and Methods

This is a pilot cross-sectional case-control (3:1) study carried out in our hospital. Subjects were recruited from hospital attendees between March 2019 and April 2021. Cases were candidates for BS (BMI > 35 kg/m^2^ with comorbidities or BMI > 40 kg/m^2^) and controls were healthy population with normal weight (BMI < 25 kg/m^2^) matched by age. The study accomplishes the STROBE guidelines for case-control studies [21]. Exclusion criteria were age over 65 years, pregnant women, patients with clinical or personal characteristics that make monitoring difficult (i.e., drug or alcohol addiction, severe psychological or psychiatric disorders) and patients with a history of trauma injury, spine injury, or any process that could affect the motor function of limbs. The hospital ethics committee approved all the procedures carried out in the study and all subjects signed the informed consent. BIA and MUS were performed on the same day.

### 2.1. BIA

The skeletal muscle mass (SM) was measured using BIA. This method measures the body composition according to the differences in electric impedance among biological tissues using Janssen’s equation: SM (kg) = [(height − 2 (cm)/BIA resistance (ohms) × 0.401) + (gender (men = 1, women = 0) × 3.825) + (age (years) × −0.071)] + 0.5102. We used the SMI of Baumgartner to identify sarcopenia in our obese population, which was calculated using SMI = SM kg/height^2^. As previously reported [1,22], those obese subjects from the lowest tertile of SMI were considered sarcopenic. 

The BodyStat^®^ 1500 MDD model was used: a dual-frequency device (5 and 50 kHz), which offers information on the resistance expressed in ohms (Ώ), as well as the percentage of FM, LM, bone mass, water content, and basal metabolic rate. The accuracy of this model is impedance: 2–3 Ω, resistance (50 kHz): +/− 2 Ω, reactance (50 kHz): +/− 1 Ω, and phase angle (50 kHz): +/− 0.2°. It uses disposable adhesive electrodes of the same brand, placed on the right hand and foot, passing the current through the trunk between the two extremities. The subjects lay supine with arms separate from the body, and legs not touching each other, the same position as the ultrasound (US) technique we described below. An excitation current of 500 a at 50 kHz was applied to the distal electrodes, and the voltage was detected by proximal electrodes.

### 2.2. Ultrasonic Technique

Ultrasound measurements were made with a sonographic US Logiq P9 (GE Healthcare) equipment muscle-skeleton B-model using a linear multifrequent transducer (4–11 Hz), with adequate use of contact gel and minimal pressure to avoid excessive compression of the muscle. Participants were positioned supine, completely relaxed, with both knees extended. The transducer was positioned perpendicular to the longitudinal axis of the quadriceps femoris to take transverse US images on the right leg, two-thirds of the distance between the anterior-superior iliac spine and the superior pole of the patella in the sagittal plane. The two tissue thickness measurements were (a) subcutaneous fat (SF): subcutaneous tissue—from the skin to the inferior border of the superficial fascia on the anterior thigh—and (b) thigh muscle thickness (TMT)—from the superior border of the rectus femoris to the inferior vastus intermedius—taking the femoral bone as a reference [23,24,25,26,27] (Figure 1). Sarcopenia mainly affects lower limbs, so the rectus femoris was specifically selected to evaluate it [28], and its US evaluation was carried out according to the recommendations of the European Union Geriatric Medicine Society Sarcopenia Special Interest Group and in accordance with the previous literature [29]. A set of three consecutive measurements was performed and the average value was reported as the TMT or SF. Data were reported in centimetres (cm) as means +/- standard deviation. To avoid interindividual variability, all measurements were performed by the same physician who has 2 years of experience. To assess intraobserver reliability, we evaluated intraclass correlation coefficients (CVs) using 3 images taken on 3 separate days on 30 participants and the results of CVs were 0.93 for TMT and 0.87 for SF.

### 2.3. Statistical Analysis

All continuous variables are expressed as mean ± standard deviation unless specified otherwise. Differences between groups were evaluated using a parametric test (chi-squared for categorical variables and *t*-test for continuous variables). The Pearson’s correlation test was used to compare BIA and US muscle and fat tissue measurements. 

The lowest tertile of SMI defined SO from the whole cohort and 9.5 kg/m^2^ set in our sample. The areas under the curve (AUC) in the receiver operating characteristic (ROC) curve were calculated to examine the relationship between TMT and SO as defined by BIA. Moreover, a cut-off value to identify SO using TMT was obtained. Youden’s index was often used in conjunction with ROC analysis. The index was defined for all points of a ROC curve, and the maximum value of the index may be used as a criterion for selecting the optimum cut-off point when a diagnostic test gives a numeric rather than a dichotomous result. Statistical analysis was performed using STATA version 14 (College Station, TX, USA) and *p* < 0.05 was considered significant. 

## 3. Results

The baseline data of the participants are shown in Table 1. A total of 122 subjects were included (cases: 90; controls: 32): 89 female participants and a mean age of 51.2 ± 9.75 years. BMI in the obesity group was 44.22 ± 5 kg/m^2^ and 24.54 ± 3.60 kg/m^2^ in the control group. The muscle and fat tissue measurements using BIA and US are shown in Table 1. TMT measurements were 1.96 cm +/− 0.76 cm in the obese group and 1.27 cm +/− 0.35 cm in the control group. The ratio between the mean of TMT and SF was 1.57 in the control group and 1.38 in the cases. Therefore, the proportion of TMT in relation to SF was significantly lower in subjects with obesity (*p* < 0.001). 

A significant correlation was observed between muscle measurement values using BIA (LM and SMI) and TMT measured by MUS (R = 0.46, *p* < 0.0001 and R = 0.47, *p* < 0.0001, respectively) (Figure 2a,b). FM assessed by BIA and SF assessed by US was significantly related (R = 0.5, *p* < 0.0001) (Figure 2c). We detected 40 patients in the lower tertile of SMI (defined as SO), all of them female. In this group, there was also a statistically significant correlation between MUS and LM (R = 0.79, *p* = 0.003). All the correlations are summarized in Table 2. 

To examine the capacity of MUS to predict SMI using BIA, we performed ROC analysis and the AUC was 0.77 (95% CI: 0.68242 to 0.84281). Using the Youden index, which gives equal weight to false positive and false negative values, the optimal cut-off point for maximum efficiency was 1.57 cm in TMT assessed by MUS (sensitivity = 75.6% and specificity = 71.1%; positive predictive value = 84.3% and negative predictive value = 58.7%). According to the ROC results, a cut-off value of 1 cm in TMT showed a sensitivity of 92.3%, and specificity of 23.7% [Figure 3]. This means that patients with obesity with a TMT less than 1 cm in the MUS have more than a 90% probability of having SO.

## 4. Discussion

The current study provides the first evidence that TMT measured by using MUS is significantly related to LM and SMI measured using BIA, a standardized method used for measuring muscle mass. Notably, this relationship persisted in the SO group. We also found a significant correlation between FM measured by BIA and SF measured by US (R = 0.5, *p* < 0.0001). In addition, we could establish a cut-off value of <1.57 cm for TMT assessed by MUS to diagnose SO in our population (sensitivity = 75.6% and specificity = 71.1%). Moreover, we observed that a TMT less than 1 cm was able to predict SO in 92.3% of cases, thus suggesting that MUS is a reliable method for screening SO in the obese population. It is a promising result for a screening test but, of course, further studies are required not only using BIA as comparison but also more accurate methods.

The latest consensus by ESPEN and EASO defends that the diagnosis of SO will be performed in two steps by sequentially assessing (a) functional parameters (strength) and (b) body composition by DXA, BIA, or CT [7]. Our results point to US as a new, simple, and useful tool to be incorporated in the study of body composition. Most of the previous studies on US measurements of skeletal muscles in lower extremities have mainly been addressed to explore the relation between muscle strength and muscle thickness [30,31]. Minetto et al. [28] suggested that MUS provides a practical and accurate tool for identifying individuals with low muscle mass. These authors assessed the relationship between the thickness of the rectus femoral muscle measured using US and the appendicular muscle mass measured using whole-body DEXA. 

The use of different definitions, thresholds, indexes, and methods to determine SM and FM have generated an extreme variability in the assessment of the prevalence of SO. Therefore, it clearly appears that there is an urgency to establish a standardized definition of this condition which will facilitate the study of its pathophysiology and medical consequences. Indeed, we know that weight gain in obese people is mainly caused by an increase in FM; however, a contemporary reduction in SM is often observed. No universally accepted validated cut-offs for SMI are currently available [7], and even the use of those proposed by the European Working Group on Sarcopenia in Older People (EWGSOP) in obese people may be misleading. For this reason, we considered in the present study that the sarcopenic group would be the tertile of our population with the lowest SMI.

The prevalence of sarcopenia and SO increases with age, and it is higher in the population undergoing BS [1,4,32,33,34]. It should be noted that obese people candidates for BS have tried multiple diets on many occasions. Caloric restriction in the context of obesity treatment is an effective strategy to achieve quick and significant weight loss; however, it causes a reduction in both FM and LM. Repeated cycles of losing-regaining weight combined with age-related changes in body composition can lead to the development of SO. Approximately 25–31% of the short-term weight loss obtained through low-calorie diets can be attributed to a loss in muscle mass [35]. Consequently, having a rapid screening test for sarcopenia in this population seems reasonable. From the therapeutic point of view, the identification of SO means that, apart from reducing body fat, strategies aimed at increasing muscle mass and strength are required [7]. 

Limitations of MRI, CT, or DXA are common when evaluating muscle mass in obesity. Most of these limitations rely on weight per se or on the difference in body composition compared to non-obese subjects and, hence, methodology is important in dealing with them. In addition, although we were aware of the patient’s weight before the visit, obese patients may feel stigmatized in a routine clinical encounter if the methodology commonly used is changed due to excess weight. This needs to be addressed to avoid stigmatization, with a clear need to make these diagnostic procedures more accessible to people with obesity [9].

The advantages of using MUS as a screening method for detecting SO are based on it being a rapid, non-expensive, non-invasive, and harmless method. In addition, it is portable, thus allowing image acquisition in every clinical setting by a doctor: community and hospital. In addition, it requires minimal training [36]. It could be argued that a portable BIA can also offer these advantages and it could be even more accurate and reproducible. However, MUS complements BIA in better phenotyping of patients with obesity, providing the concept of regional sarcopenia by a specific muscle group and is not affected by the factors like extreme BMI or hypervolemia [37]. In fact, a patient could have BIA results without sarcopenia but present sarcopenia in a specific muscle group, being that the quadriceps are a very important muscle for a person’s autonomy. Specifically, it has been demonstrated in several studies that MUS measurement of the thickness of quadriceps femoris is highly reproducible in sarcopenic patients [38,39]. Moreover, it correlates with isometric maximum voluntary contraction force, probably due to the higher spatial resolution of this large muscle [12]. Furthermore, the same probe can be used for thyroid US, hence facilitating its use in the endocrinology units. Finally, the role of MUS in monitoring the effect of different treatments for obesity remains to be elucidated but could open a new clinical research area in the field of personalized medicine [40]. 

The disadvantages of US are principally the lack of standardization and its high dependence on the expertise and skills of the operator [41]. The interpretation of muscle–fat interfaces is limited due to similar acoustic impedance of muscle and fat tissues. Another drawback of the US technique is that the operator could cause measurement errors by applying the transducer to the skin with excessive pressure because this may compress the muscle [42].

The present study has several limitations. First, the MUS images were only taken at three random sites on the anterior thigh. As a result, this may not be representative of either the whole thigh muscle or the lean-body mass. However, a good correlation with both LM and SMI assessed by BIA was observed. Multiple site measurements should be examined and counterbalanced with the actual time consumed to see whether they could provide a better representation of whole-body sarcopenia. Second, we did not have data on muscle function. However, given that the quadriceps is an important muscle for mobility measurement, quadriceps thickness provides a useful surrogate of force [43,44,45]. Thirdly, all measurements were performed by the same physician, and this precluded the test reproducibility. Finally, this is a pilot study and, therefore, larger studies are needed to confirm our results, mainly because a significant error with respect to gold standard measurements could be present.

## 5. Conclusions

In conclusion, this pilot study suggests that morphological characteristics based on US measures of the quadriceps muscle can be used for the screening and initial evaluation of SO in candidates for BS. Larger studies to validate this simple low-cost screening strategy are warranted. 

## Figures and Tables

**Figure 1 jcm-11-02886-f001:**
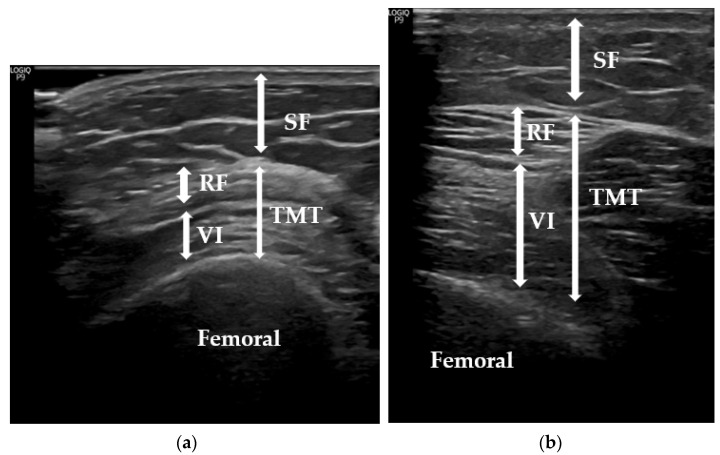
Measurement of subcutaneous tissue and thigh muscles using US. SF: subcutaneous fat; TMT: thigh muscle thickness; VI: vastus intermedius; RF: rectus femoris. (**a**) Representative image of a participant control of our sample; (**b**) representative image of a participant case (candidate for BS) of our sample.

**Figure 2 jcm-11-02886-f002:**
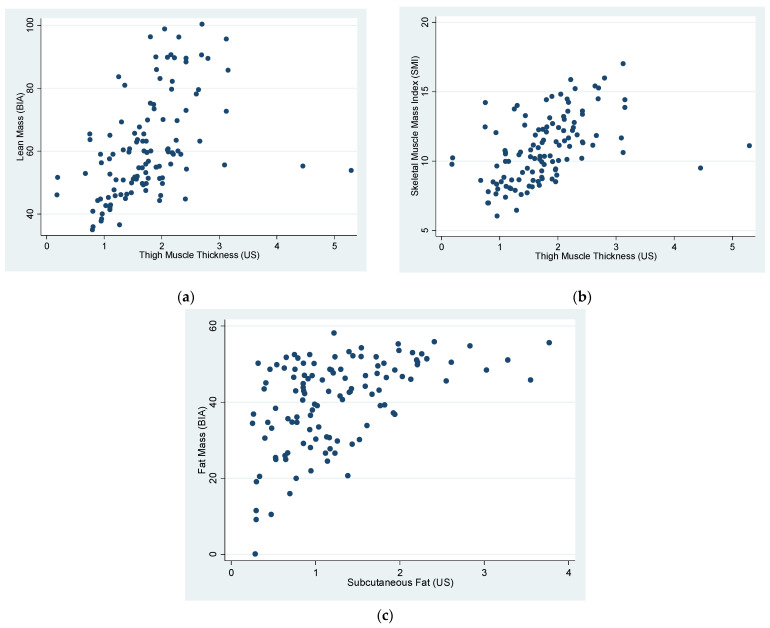
(**a**) A significant correlation was observed between lean mass (LM) by bioelectrical impedance analysis (BIA) and thigh muscle thickness (TMT) by ultrasound (US); (**b**) a significant correlation was observed between skeletal muscle mass index (SMI) assessed by bioelectrical impedance analysis (BIA) and thigh muscle thickness (TMT) assessed by ultrasound; (**c**) a significant correlation was observed between fat mass (FM) assessed by bioelectrical impedance analysis (BIA) and subcutaneous fat (SF) assessed by ultrasound (US).

**Figure 3 jcm-11-02886-f003:**
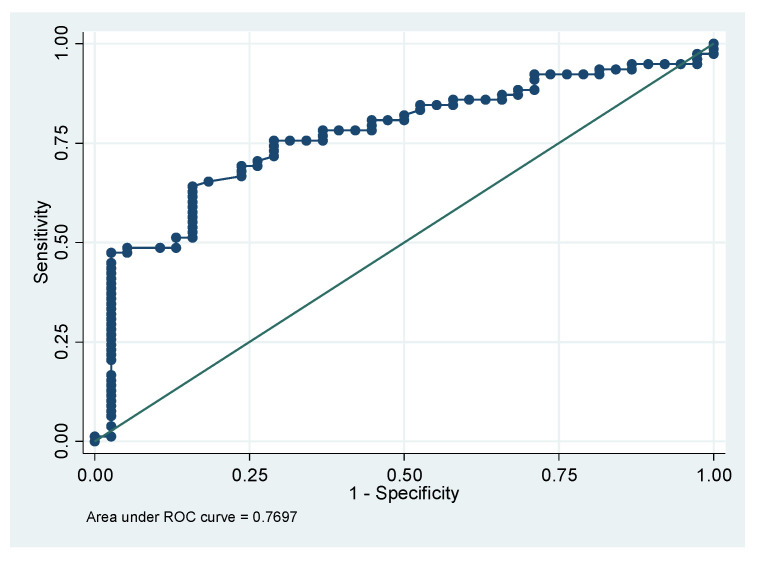
To assess the capacity of muscle ultrasound (MUS) to predict skeletal muscle index (SMI) using bioelectrical impedance analysis (BIA), we performed ROC analysis and the AUC was 0.77.

**Table 1 jcm-11-02886-t001:** Basal characteristics of participants and muscle and fat tissues measurements using BIA and US.

	Total	BS-Group	C-Group	*p* Value
**Number of participants**	122	90	32	-
**Age (years)**	51.2 ± 9.75	52.02 ± 9.40	49 ± 10.53	-
**Female**	89 (73%)	64 (71%)	25 (78%)	-
**BMI (kg/m^2^)**	39.05 ± 9.86	44.22 ± 5	24.54 ± 3.60	<0.001
** *BIA* **				
**Lean mass (%)**	61.15 ± 16.03	65.44 ± 15.36	49.36 ± 11.33	<0.001
**Fat mass (%)**	40.43 ± 11.70	45.61 ± 7.28	26.53 ± 9.88	<0.001
**SMI (kg/heigh^2^)**	10.84 ± 2.35	11.58 ± 2.14	8.81 ± 1.63	<0.001
** *US* **				
**TMT (cm)**	1.77 ± 0.74	1.96 ± 0.76	1.27 ± 0.35	<0.001
**SF (cm)**	1.25 ± 0.72	1.42 ± 0.74	0.81 ± 0.39	<0.001

Values are expressed as mean ± standard deviation and n (%). BIA: Bioelectrical impedance analysis; US: ultrasound; SMI: skeletal muscle mass index; TMT: thigh muscle thickness, SF: subcutaneous.

**Table 2 jcm-11-02886-t002:** Relationship between LM (BIA), TMT (US), and SMI (BIA.

*Correlations*	R	*p*
**LM (BIA) vs. TMT (US)**	**0.46**	**<0.0001**
**LM (BIA) vs. TMT (US)** **(Sarcopenic group)**	**0.79**	**0.0033**
**SMI (SM kg/ height^2^ m^2^) vs. TMT (US)**	**0.47**	**<0.0001**
**FM (BIA) vs. SF (US)**	**0.5**	**<0.0001**

LM: lean mass; BIA: Bioelectrical impedance analysis; TMT: thigh muscle thickness; US: ultrasound; SMI: skeletal muscle mass index; FM: fat mass, SF: subcutaneous.

## Data Availability

Not applicable.

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
