# Peer review of "Usefulness of Muscle Ultrasound to Study Sarcopenic Obesity: A Pilot Case-Control Study"

_jcm, 2022, doi:10.3390/jcm11102886_

Round 1
Reviewer 1 Report
This is an interesting paper on how to assess sarcopenic obesity with a simple tool (US).
In my opinion Authors should better explain and mention references to clarify why they chose thigh muscles, namely quadriceps femoris and excluded other muscle (e.g. paraspinal or psoas) usually employed to assess sarcopenia with CT/MRI.
As there are plenty of abbreviations throughout the text, I would suggest to pool them up before introduction, to make manuscript consultation easier.
Reviewer 2 Report
The original manuscript was an ambitious attempt to establish a cutoff value to assess sarcopenic obesity using ultrasound. This attempt is in line with the “Endocrinology & Metabolism” section of the special issue. The substantial strength of this research is not remarkable. Moreover, it falls short in a lot of ways.
For example, in the case of mice, it is possible to separate the weight of its fat and muscle volume by dissect. But all human body composition evaluation methods are indirect estimation methods. Such like underwater weighing method, MRI, stable isotope dilution method, etc. are indirect methods based on basic theories such as “Physics”, “Chemistry”. On the other hand, the BIA method is a secondary indirect method which validity has been confirmed by an indirect method using basic theory such as the MRI method. In this study, using US is evaluated BIA as a valid standard, so the error value is larger.
US method is highly dependent on the measure’s skill. There are two problems. First, as a research, there is an error in the measured value depending on who measures it by US method. Therefore in this study, the same physician measured all of them. However, the current manuscript is not possible to confirm the accuracy and reproducibility. The author needs to describe the individual error of the physician (ICC and CV). The author also needs to describe how to measure by US method. For example, transverse images, using water-soluble transmission gel, using B mode, contact with or without compressing the skin surface, TMT and FT were measured by using electric caliper on frozen image or other methods, how many times measuring, and so on.
Second, as a clinical settings, the measurer must be high skill for using US to evaluate subject. Using MRI or BIA, etc, might be easy to consider generalization on clinical settings. The author should explain the advantages and dis advantages of the US method carefully. In addition, there might be countries where US method for human is possible only to used by healthcare professional such as doctors. Nevertheless, is it better to use the US method. If it was reported such as “the BIA method is affected by postoperative edema, but the US method is not affected by edema”, it is clinical advantage.
Round 2
Reviewer 2 Report
It was successfully corrected some of the manuscripts, but there are some inherently inadequate parts.
In the first review, I pointed out that BIA method is a secondary indirect method. In this study, using US is evaluated BIA as a valid standard, so the error value is larger. If using the present method, it is necessary to carefully describe what kind of systematic error there is and what method was used to suppress the effect of the error. For example, there is no description of the accuracy of The BodyStat® 1500 MDD model. How much R2 score for The BodyStat® 1500 MDD model based on MRI, DXA, stable isotope dilution, etc.? Furthermore, what is the expected margin of error between The BodyStat® 1500 MDD model of considering accuracy based on MRI or something and US in this study? Is the range of error obtained by the calculation accurate enough to estimate muscle mass in US, or is it only possible to evaluate the order in US? There are some systematic errors to consider besides The BodyStat® 1500 MDD model. It is necessary to describe how to inhibit the effects of each possible systematic error.
With a portable BIA tool such as the InBody S10, it is possible to measure short amount of time. In order to measure the thigh by the US, it is necessary to prepare in advance such as exposing the thigh. In addition, as authors have mentioned, the US depends on the skill of the measurer. US is not possible to be considered as a useful tool compared to BIA tools. It would be faster, easier, more objective and reproducible to identify SO directly by the BIA rather than by US.
Thank you for calculate Intra-rater reliability (ICC[1,1]).
Author Response
It was successfully corrected some of the manuscripts, but there are some inherently inadequate parts.
Thank you for the second review process of our paper. We have tried to answer all your queries and we hope that you will find now the new revised manuscript suitable for publication.
In the first review, I pointed out that BIA method is a secondary indirect method. In this study, using US is evaluated BIA as a valid standard, so the error value is larger. If using the present method, it is necessary to carefully describe what kind of systematic error there is and what method was used to suppress the effect of the error. For example, there is no description of the accuracy of The BodyStat® 1500 MDD model. How much R2 score for The BodyStat® 1500 MDD model based on MRI, DXA, stable isotope dilution, etc.? Furthermore, what is the expected margin of error between The BodyStat® 1500 MDD model of considering accuracy based on MRI or something and US in this study? Is the range of error obtained by the calculation accurate enough to estimate muscle mass in US, or is it only possible to evaluate the order in US? There are some systematic errors to consider besides The BodyStat® 1500 MDD model. It is necessary to describe how to inhibit the effects of each possible systematic error.
As required, some details regarding the error of The BodyStat® 1500 MDD model have been added to the Material and Methods section of the revised manuscript. It should be noted that in this paper we are just proposing the tight-US as screening test to identify those patients who would need gold standard techniques to assess sarcopenia. In addition, we are not considering BIA as a valid standard but a “conventional method” (please see last paragraph of the introduction). In this regard, in the last systematic review to define sarcopenic obesity, 21 out of 75 studies used the BIA for this purpose (Donini et al. Clin Nutr. 2020;39:2368-2388. PMID: 31813698). Finally, we would like to emphasize that the nature of our paper (case-control study) lead to homogenize the errors in both groups, thus making our results useful for comparisons and to select a valid cut-off for screening. Nevertheless, we fully agree with the referee that a significant error respect to gold standard measurements could be present and, therefore, we have added this important limiting factor at the end of the discussion section of the revised manuscript.
With a portable BIA tool such as the InBody S10, it is possible to measure short amount of time. In order to measure the thigh by the US, it is necessary to prepare in advance such as exposing the thigh. In addition, as authors have mentioned, the US depends on the skill of the measurer. US is not possible to be considered as a useful tool compared to BIA tools. It would be faster, easier, more objective and reproducible to identify SO directly by the BIA rather than by US.
The referee is right in indicating that a portable BIA could be even more accurate and reproducible than US in identifying sarcopenic obesity. However, in the setting of Endocrinology Departments, which are receiving a high number of obese patients to be evaluated, the advantage of using US is clear because the same probe currently used for thyroid US can also be used for the screening of sarcopenic obesity. In addition, US could complement BIA in better phenotyping of patients with obesity, providing the morphological status of a key muscle such is the quadriceps and avoiding the interference of confounding factors like extreme BMI or hypervolemia (Sengul Aycicek et al. Acta Clin Belg. 2021, 76, 204-208). Furthermore, a useful information on functionality can be extrapolated. In this regard, it has been shown that thickness of quadriceps femoris correlates with isometric maximum voluntary contraction force (Rustani al. Arch Gerontol Geriatr. 2019, 83:151-154). A new paragraph explaining this important issue raised by the referee has been added to the revised manuscript.
Thank you for calculate Intra-rater reliability (ICC[1,1]).
The methodology with such information is actually more precise. Thank you very much.
